# Neural and Onconeural Autoantibodies and Blood–Brain Barrier Disruption Markers in Patients Undergoing Radiotherapy for High-Grade Primary Brain Tumour

**DOI:** 10.3390/diagnostics14030307

**Published:** 2024-01-31

**Authors:** Katarzyna Hojan, Krystyna Adamska, Agnieszka Lewandowska, Danuta Procyk, Ewa Leporowska, Krystyna Osztynowicz, Slawomir Michalak

**Affiliations:** 1Department of Occupational Therapy, Poznan University of Medical Sciences, 61-781 Poznan, Poland; 2Department of Rehabilitation, Greater Poland Cancer Centre, 61-866 Poznan, Poland; 3Department of Radiotherapy, Greater Poland Cancer Centre, 61-866 Poznan, Poland; krystyna.adamska@wco.pl (K.A.); agnieszka.lewandowska@wco.pl (A.L.); 4Department of Elektroradiology, Poznan University of Medical Sciences, 61-701 Poznan, Poland; 5Laboratory Ward, Greater Poland Cancer Centre, 61-866 Poznan, Poland; danuta.procyk@wco.pl (D.P.); ewa.leporowska@wco.pl (E.L.); 6Department of Neurochemistry and Neuropathology, Neurology Department, Poznan University of Medical Sciences, 60-355 Poznan, Poland; osztynowiczkr@ump.edu.pl (K.O.); swami@ump.edu.pl (S.M.); 7Department of Neurosurgery and Neurotraumatology, Poznan University of Medical Sciences, 60-355 Poznan, Poland

**Keywords:** antibodies, glioma, immunology, radiation, cancer

## Abstract

Radiotherapy (RT) plays a key role in brain tumours but can negatively impact functional outcomes and quality of life. The aim of this study was to analyse anti-neural and onconeural autoantibodies and markers of blood–brain barrier (BBB) disruption in patients with primary brain cancer undergoing RT. Materials and methods. A prospective study was conducted on 45 patients with a brain tumour scheduled for intensity-modulated radiotherapy. Assessments were performed at baseline, post-RT, and at three months. We measured serum levels of BBB disruption biomarkers and anti-neural, onconeural, and organ-specific antibodies. Results. Antibodies against nucleosome antigens and neuronal surface antigens were detected in 85% and 3% of cases, respectively; anti-neural and onconeural antibodies were observed in 47% and 5.8%. In 44% patients, ≥2 antibody types were detected. No significant changes in BBB biomarkers were observed. Conclusion. The findings of this study show that a humoral immune response is common in patients undergoing RT for brain cancer. This response appears to be non-organ specific but rather directed against nucleosome antigens, but onconeural antibodies were uncommon, suggesting a low risk of a neurological paraneoplastic syndrome. Our data suggested that radiotherapy may not affect BBB integrity, but larger studies are needed to better characterise the pathophysiological effects of RT.

## 1. Introduction

Every year, thousands of patients around the world undergo radiotherapy (RT) for the treatment of a primary brain tumour. Despite the important role that RT plays in the treatment of these patients, the brain is a highly sensitive and crucial organ that is highly susceptible to radiation damage. Radiation-induced injury is multifactorial and complex, mainly characterised by vascular abnormalities, inflammation, gliosis, demyelination, and—at high doses—white matter necrosis [1,2]. Standard radiation treatment for brain tumours includes high dosage megavoltage radiation to the cranial vault; however, a high proportion (50% to 90%) of these patients develop impaired cognition and dysfunction, which is highly disabling and progressive [1,3,4]. 

Damage to the blood–brain barrier (BBB), either due to the neoplasm itself or to radiotherapy, may play an important pathogenic role by permitting the passage of immunoglobulins or other agents [5,6,7]. Similarly, a damaged BBB may be associated with tumour growth and cognitive and/or behavioural impairment after surgery or RT. These alterations to the BBB can provide valuable information, thus providing a useful surveillance tool [5,6,7,8]. The main role of the BBB is to protect the brain and to maintain homeostasis within the central nervous system (CNS). This protection is achieved through a unique microvasculature network that physically blocks and selectively transports specific molecules across the barrier via the combined efforts of endothelial cells, tight junction (TJ) proteins, basal lamina, and astrocytes that make up the neurovascular unit [6,7]. The TJs are composed of transmembrane proteins (claudins and occludins) and scaffolding proteins (zonula occludens-1 protein) that anchor their branches to the actin cytoskeleton [7]. Several studies have shown that circulating TJs are clinically significant biomarkers of BBB disruption [8,9,10]. In many cerebrovascular diseases, the breakdown of the BBB is common and characterised by the infiltration of blood components and an altered transport and clearance of molecules into the CNS [6,7]. For example, persistently increased levels of S100β, a calcium-binding protein commonly found in astrocytes [11], indicates the continuous release of this protein from damaged tissues. Elevated serum levels of S100β correlate with neurological deterioration after cardiac surgery [12,13] and have also been associated with a poor likelihood of survival after hypoxia [14]. S100β levels are a recognized marker of traumatic brain injury [15,16,17] and BBB dysfunction, even in the absence of apparent brain injury [18]. Studies have found that serum S100β levels are associated with poor neurological outcomes in a range of clinical scenarios, including surgical insult to the CNS [19], operative decompression of cord metastases [20], and brain metastases [21,22]. Kanner et al. [18] found that S100β levels were associated with the development of cognitive changes after brain injury. Given these findings, it seems logical to hypothesise that RT could disrupt the BBB in patients undergoing RT for brain cancer. Consequently, evaluating these markers could provide useful data about the functional status of these patients.

Anti-neuroendothelium antibodies may interact with endothelial cells located in the central and peripheral nervous systems, mainly the *vasa nervorum*. Some authors [23] have suggested that onconeural antibody levels and clinical symptoms during oncological treatment may correlate with BBB integrity. Koszewicz et al. evaluated a wide range of markers in a series of brain cancer patients and found the presence of several types of autoantibodies, including anti-endothelium, anti-GFAP (glial fibrillary acidic protein), and antinucleosome antibodies [23]. However, they did not detect the presence of onconeural antibodies. To our knowledge, the possible correlations between circulating antibodies (onconeural, anti-neural) and physical impairment in patients with brain tumours undergoing RT have not been sufficiently evaluated to date. 

We hypothesised that previously undetected and comparatively subtle early manifestations of radiation-induced damage to the CNS may synergize over time to form long-term macro- and microstructural abnormalities such as BBB disruption due to the presence of anti-neural and/or onconeural antibodies. However, due to the lack of data, it is not clear whether the disruption of the BBB and the accompanying rise in serum S100β causes an immune response leading to the production of these antibodies. 

Therefore, in the present study, we sought to measure various different types of antibodies and then to determine whether they associate with measures of clinical function. We also sought to evaluate the immunological role of specific molecular markers in patients following the completion of RT.

In short, the main aim of this study was to analyse anti-neural and onconeural autoantibodies and biomarkers of BBB disruption in patients undergoing RT for primary brain tumours. 

## 2. Materials and Methods

### 2.1. Patients

Patients were recruited from the Department of Radiotherapy at a cancer centre between 1 April 2022 and 31 May 2023. The study protocol was approved by the Internal Review Ethics Board (Approval code: 703/18) at the Poznan University of Medical Sciences and registered at ClinicalTrials.gov (identifier: NCT05192447). All participants provided written informed consent to participate in this study. 

Inclusion criteria were as follows: diagnosis of a primary brain tumour; scheduled to undergo RT; age: 18 to 70 years; good general health condition (Eastern Cooperative Oncology Group [ECOG] score = 0–2); and written informed consent. 

Exclusion criteria were as follows: ≥two tumours or metastases; presence of psychological or psychiatric illness under pharmacotherapy; other neurological disorders (e.g., stroke, multiple sclerosis, Parkinson’s disease, meningitis, etc.); autoimmune disease (autoimmune encephalitis, Hashimoto encephalopathy, etc.); or clinically significant circulatory failure. 

### 2.2. Study Design

This study was a single-centre prospective clinical trial. Assessments were performed at three time points: at baseline (one day prior to the start of RT [T0]), one day after completion of RT (T1), and 3 months after completion of RT (T2). 

### 2.3. Radiotherapy

All patients underwent intensity-modulated radiotherapy (IMRT) that allows the radiation dose to conform more precisely to the three-dimensional shape of the tumour by changing—modulating—the radiation beam into multiple smaller beams. This enables a higher dose of radiation to be delivered to the tumour while sparing healthy tissue around it. IMRT uses linear accelerators to safely deliver precise radiation to a tumour while minimizing the dose to surrounding normal tissue. Our patients were treated by IMRT using a conventional fractionation scheme (2 Gy per dose, total dose = 60 Gy) administered over a 30 day period following the schedule described by Scaringi et al. [24]. 

### 2.4. Laboratory Assay

Blood was drawn at the hospital laboratory in the morning under fasted conditions (i.e., before breakfast). All blood samples were frozen to −80 °C. The serum biomarkers were evaluated at the Department of Radiobiology in collaboration with the Department of Neurochemistry and Neuropathology at the Poznan University of Medical Sciences. 

#### 2.4.1. BBB Biomarkers

Markers of BBB integrity were assessed by measuring serum concentrations of S100β, occludin (OCLN), claudin-5 (CLN5), and zonula occludens 1 (Zo-1). To estimate OCLN concentrations, rabbit and mouse anti-human OCLN antibodies (Zymed Labolatories Inc, South San Francisco, CA, USA) were used as capture and detection antibodies, respectively. CLDN5 concentrations were estimated with an in-house enzyme-linked immunosorbent assay (ELISA), using mouse anti-human CLDN5 (Zymed Labolatories Inc.) and rabbit anti-human CLDN5 (Abcam Inc., Cambridge, UK) antibodies for capture and detection, respectively. Zo-1 levels were analysed with rabbit anti-human Zo-1 and mouse anti-human Zo-1 (Invitrogen, Waltham, MA, USA) antibodies for capture and detection, respectively. Goat anti-mouse IgG (H+L)−HRPO was used as the secondary antibody for the OCLN and Zo-1 ELISAs, and goat anti-rabbit IgG (H+L)−HRPO (Invitrogen) was used for the CLDN5 ELISA. 

S100β levels were determined using commercially available ELISA kits according to the manufacturer’s instructions (BioVendor Laboratory Medicine Inc., Brno, Czech Republic; and R&D Systems Inc., Minneapolis, MN, USA). 

#### 2.4.2. Analysis of Antibodies

We evaluated the following types of antibodies: (1) onconeural antibodies (anti-Hu, anti-Ri, anti-Yo, anti-Ma/Ta, anti-Cv2, and anti-amphiphysin) by indirect immunofluorescence confirmed with line blot with the use of recombinant antigens (EUROIMMUN, Lubeck, Germany); (2) anti-neural antibodies (anti-myelin, anti-myelin-associated glycoprotein [anti-MAG], and anti-glutamic acid decarboxylase [anti-GAD]) by indirect immunofluorescence only (EUROIMMUN, Lubeck, Germany); and (3) superficial anti-neuronal antibodies (anti-NMDA, -AMPA, -GABA, -LGI, and -CASPR) in serum by cell-based assay (CBA; EUROIMMUN, Lubeck, Germany). The Department of Neurochemistry and Neuropathology in Poznan participates in the international system of quality control and systematically obtains quality certificates from the Institut für Qualitätssicherung (Kiel, Germany).

### 2.5. Functional Assessment

The Functional Assessment of Cancer Therapy-General (FACT-G) questionnaire and the Functional Independence Measures (FIM) system were used to assess the participants’ physical, psychological, and social status [25,26]. 

The 27-item FACT-G scale assesses functional status and quality of life (QoL) in four domains: Physical Well-Being (PWB), Emotional Well-Being (EWB), Social/Family Well-Being (SWB), and Functional Well-Being (FWB). The highest possible total score on the FACT-G is 108 points, with higher scores indicating better QoL and functional status [26,27]. 

The FIM is an 18-item multidimensional scale designed to evaluate physical, psychological, and social function in patients with neurological illnesses [28,29,30]. It assesses performance in six areas (self-care, continence, mobility, transfers, communication, and cognition). The scale assesses the patient’s degree of dependence on the help of others to perform everyday activities. The FIM is commonly used to assess the degree of disability and changes in response to rehabilitation or medical interventions [29,30]. 

For the cognitive assessment, we used Addenbrooke’s Cognitive Examination III (ACE III), which includes tests to assess attention, orientation, memory, language, visual perception, and visual–spatial skills. The ACE III has proven effective in measuring cognitive impairment [31,32].

### 2.6. Statistical Analysis

For determination of the sample size, we assumed a test power of 80% with a cut-off for statistical significance of *p* = 0.05. To calculate the effect size, we used data from previous publications [9,23,25,33] describing the relationship between markers of BBB disruption in other types of cancer and set the effect size at 0.5. Given these assumptions, the minimum sample size was 28 participants.

Statistica StatSoft, v.13.1 (StatSoft, Inc. 2016, STATISTICA, Kraków, Poland) was used to perform the data analysis. The Shapiro–Wilk tests [34] were used to determine distribution normality. For variables with a normal distribution, repeated measures ANOVA and Tukey’s post hoc test were used to compare changes over time (T0, T1, T2). In the absence of a normal distribution and for ordinal variables, the Friedman test with Dunn’s post hoc test was used. The Mann–Whitney test was used to compare two groups. Additionally, Spearman’s r correlation coefficient was determined to measure the strength and direction of the association between the two ranked variables (antibodies and BBB markers). The threshold for significance was set at 0.05.

## 3. Results

### 3.1. Characteristics of the Study Group

Forty-five patients were initially enrolled in the study. However, eleven were excluded from the final analysis for the following reasons: four withdrew consent during the observation phase, five developed cancer progression, one was diagnosed with new-onset psychiatric illness, and one died. Therefore, a total of thirty-four patients were included in the final analysis (after three months of RT). Table 1 shows the characteristics of the study group.

### 3.2. Onconeural and Anti-Neural Antibodies

In our study, onconeural antibodies (anti-CV2 and anti-Yo) were detected in two (5.9%) patients, anti-neural antibodies (anti-GAD, anti-neuroendothelium, anti-GFAP) in sixteen (47.05%) cases, antibodies against nucleosome antigens in twenty-eight (82.3%) patients, and antibodies against neuronal surface antigens (anti-CASPR) in one case (2.9%). In fifteen (44.1%) patients, ≥two types of antibodies were observed. 

### 3.3. Blood–Brain Barrier Biomarkers

Table 2 shows changes in TJ proteins and S100β during the study (time points T0, T1, and T2). As the table shows, no statistically significant changes were observed in the levels of BBB markers. 

### 3.4. Functional Assessment Results

There were no significant differences over the study duration in the analysed total scores (FACT-G, FIM, ACE III). No important (*p* < 0.05) changes were noticed in FACT-G subscores following our observations. Statistically significant decreases in functional status (FIM scale) were observed on various subscales (self-care, social awareness, and communication) at three months post-RT. Attention (measured by ACE III) decreased significantly (*p* < 0.05) following the completion of RT, as evidenced by the decrease in median scores on that subscale. No other significant changes in functional measurements were observed. Table 3 presents the functional status results (FACT-G, FIM, and ACE III) at each time point. Total scores and subscores are shown.

### 3.5. Comparison between Immunological Status and Clinical Parameters in Patients’ Subgroups

To analyse changes in patients’ functioning during the study, we observed changes between two subgroups of BT patients. 

#### 3.5.1. Antibodies and BBB Markers

We did not notice any important (*p* < 0.05) differences in the levels of selected BBB markers over the duration of the study between the analysed subgroups of patients (with one type of autoantibody or with onconeural antibodies vs. patients with a negative humoral response to them). 

#### 3.5.2. Autoantibodies and Functional Status of BT Patients

At first, we observed patients with positive immunological responses—with autoantibodies vs. without them. In most cases, the autoantibodies did not connect significantly with clinical measurements of functional status (*p* > 0.05). At the T2 assessment, median scores on the FACT-G PWB subscale were statistically significantly higher in patients with a humoral immune response who produced any of the tested autoantibodies compared to seronegative patients. Next, we analysed changes in patients with onconeural antibodies vs. functional status. At T0, we noticed statistically significant differences between patients with a negative response of onconeural antibodies and the subgroup that produced those antibodies in terms of FACT and the ACE III language subscale. According to the comparison of the FIM locomotion subscale between patients who did not have onconeural antibodies and patients with those antibodies over the study duration shown at T1, median (IQR) scores on the FIM locomotion subscale were higher in patients without antibodies than in the group with onconeural antibodies, with 14 (14) vs. 13 (12–14), respectively (*p* = 0.002). Additionally, we noticed statistically important changes in the FIM total score from T0 (*p* < 0.001) to T1 (*p* = 0.018) in analysed patient subgroups. After the end of RT (at T1), we found median (IQR) scores to be significantly lower in patients with more than one type of analysed antibodies than in patients who presented a negative humoral response on the following FIM subscales: in FIM social awareness, 17.5 (15–20) vs. 21 (16–21), respectively (*p* = 0.024); and in FIM locomotion, 13.5 (12–14) vs. 14 (14–14) (*p* < 0.001). Additionally, the FIM locomotion subscale differed significantly (*p* = 0.008) after three months of RT (T2): 9 (6–12) vs. 14 (6–14) (patients with minimum two types of antibodies vs. patients without them). 

Figure 1 shows selected functional scales and subscores during study time between two different study subgroups (with or without different antibodies).

### 3.6. Correlations between Blood–Brain Barrier Markers and Autoantibodies

We determined correlations between biomarkers of BBB damage (OCLN, CLN5, Zo-1, and S100β) and the production of autoantibodies. However, no significant correlations (*p* > 0.05) were detected at any time point. 

## 4. Discussion

Despite the importance of cognitive and behavioural impairments following the radiation therapy of the CNS, these aspects remain mostly unexplored [2,4,34,35,36,37]; thus, the mechanisms underlying these impairments are not well understood [38,39]. Therefore, we measured the molecular indicators of BBB disruption and tested for the presence of key onconeural and other types of autoantibodies to determine whether these were correlated with functional status in patients undergoing RT for brain cancer. Our results showed that 47.05% of the patients presented an immune reaction against nervous system antigens. However, onconeural antibodies were detected in only 6% of patients, suggesting a low risk of developing a neurological paraneoplastic syndrome. TJ proteins are the main components of tight junctions between the brain endothelial cells that are crucial to maintaining BBB integrity. Previous studies have shown that these proteins are present in patients with BBB disruption [40,41,42,43]. One study [43] found that OCLN expression in the brain vascular endothelium was significantly higher than in the non-nervous tissue vascular endothelium, which explains why BBB permeability is significantly lower than in other blood tissue barriers. Although some studies have shown an association between downregulated or degraded OCLN and damaged BBB function [42,43], these markers of BBB disruption have not been previously evaluated in the context of radiotherapy (and RT-related side effects) in patients with brain tumours. Importantly, in our study, we did not observe any significant changes in these markers of altered BBB function. 

One of the key aims of this study was to determine whether BBB disruption and the accompanying surges of the astrocytic protein S100β or TJs in blood cause an immune response (i.e., the production of certain antibodies). Previous studies showed that not only tissue-level examination but also serum concentrations of TJs can indicate BBB disruption and can be used as clinically useful biomarkers on a spectrum of central nervous system pathologies like ischemic stroke [8], foetal growth restriction syndrome, and multiple sclerosis [9,10]. We tested for a range of different onconeural antibodies, anti-neural antibodies, and superficial anti-neuronal antibodies. We did not notice important changes in BBB markers in BT patients who had a positive immune response. Finally, we observed correlations between analysed BBB biomarkers and antibodies. We did not detect significant correlations (*p* > 0.05) at any time point. These observations are opposite to those of other authors [23].

To our knowledge, this is the first study in patients with brain tumours which aims to determine the association between circulating antibodies (onconeural and anti-neural) and cognitive, physical, or paraneoplastic syndromes. 

In a previous study, Koszewicz et al. [23] found that anti-neuroendothelium, anti-GFAP, anti-MAG, anti-PCNA, and anti-Ro52 antibodies were associated with peripheral nerve alterations in patients with brain tumours. In that study, significantly higher vibratory and pain thresholds for cold and warmth in the upper and lower limbs were observed in the experimental group compared to controls; in addition, more than 15% of patients in the experimental group (33 patients with newly diagnosed brain cancers) tested positive for antibodies within 2–4 days after hospital admission. The functional status of our patients with BT was assessed using widely known functional scales for physical and cognitive measurements in neurological dysfunction syndromes [25,26,27,28,29,30,31,32].

Although immune reactions against systemic antigens are closely associated with disease, in our study, only 6% of cases showed onconeural symptoms, a finding that suggests a low risk of developing a neurological paraneoplastic syndrome. It should be noted that this BT subgroup, with onconeural antibodies, presented lower physical well-being on the FACT-G scale, locomotion on the FIM scale, and language on the ACE III scale during the study. In our clinical observations, we detected anti-neural antibodies in 47% of study participants, antibodies against nucleosome antigens in 82.3%, and antibodies against neuronal surface antigens in 2.9%.

Despite the high prevalence of the first two antibodies, we did observe any selected important changes in the functional status of the study participants. At first, FIM total score was statistically significantly higher in patients without humoral response than those who had positive autoantigen tests (more than two types of autoantigens). Based on the FIM scores, the functional status decreased significantly over the course of the study. FIM subscores (self-care, awareness, and communication) all decreased significantly at the three-month follow-up (T2). These findings contrast with other reports which have not found any decreases in these scores after RT [44,45]. However, our analysis of immune response in patient subgroups showed that after the end of RT (at T1) the FIM awareness and locomotion subscale scores were significantly lower (in point levels) in patients who had more than one type of analysed autoantibody compared to BT patients who presented a negative humoral response. Additionally, the FIM locomotion subscale score was significantly lower after three months of RT. ACE III scores were largely unchanged after RT, except for a significant worsening on the attention subscale, a finding that is consistent with previous reports [46,47]. We observed some significant changes in FACT-G scores during radiation treatment. FACT-G PWB subscores were importantly higher in patients with anti-neural antibodies at three months post-RT (T2); the same FACT-G subscores were significantly higher in patients who exhibited a humoral response (presence of autoantibodies) than those who were seronegative. To our knowledge, this is the first time that such clinical observations have been reported in patients with BT.

The data obtained because this study are valuable because they expand our general understanding of the potential association between markers of neuronal lesions (S100β), TJ/BBB stability, immune response (as measured by onconeural, anti-neural, and/or anti-nucleosome antibodies), and functional outcomes in the brain cancer patient population. Importantly, we did not observe any progression in the markers of BBB damage during this study.

These data may prove valuable in the development of knowledge in oncology and could also contribute to future clinical protocols for the assessment of cognitive and physical functions, thus helping to identify ways to treat deficits in patients undergoing CNS radiation therapy.

### Strength and Limitations of Study

The main limitation of this study, which can be considered a pilot study, is the small sample size (*n* = 34). By contrast, a major strength of the study is that it is one of the first to analyse BBB integrity and autoimmunity in patients with primary brain tumours who have undergone radiotherapy. In addition, this is the first clinical study to provide data on serum concentrations of OCLN, CLN5, and Zo-1 in patients with brain tumours during IMRT. To better understand the functional analysis of BT patients during RT, a multivariate analysis should be performed.

## 5. Conclusions

The findings of this study show that a humoral immune response is common in patients undergoing RT for primary brain cancer. This response appears to be non-organ specific but rather directed against nucleosome antigens. Importantly, our data suggest that RT may not affect BBB integrity. To better understand pathophysiology in patients with brain tumours during radiation treatment and to select potential therapeutic targets, larger studies are needed to determine the factors driving BBB damage, the production of autoimmune biomarkers, and their association with functional outcomes.

## Figures and Tables

**Figure 1 diagnostics-14-00307-f001:**
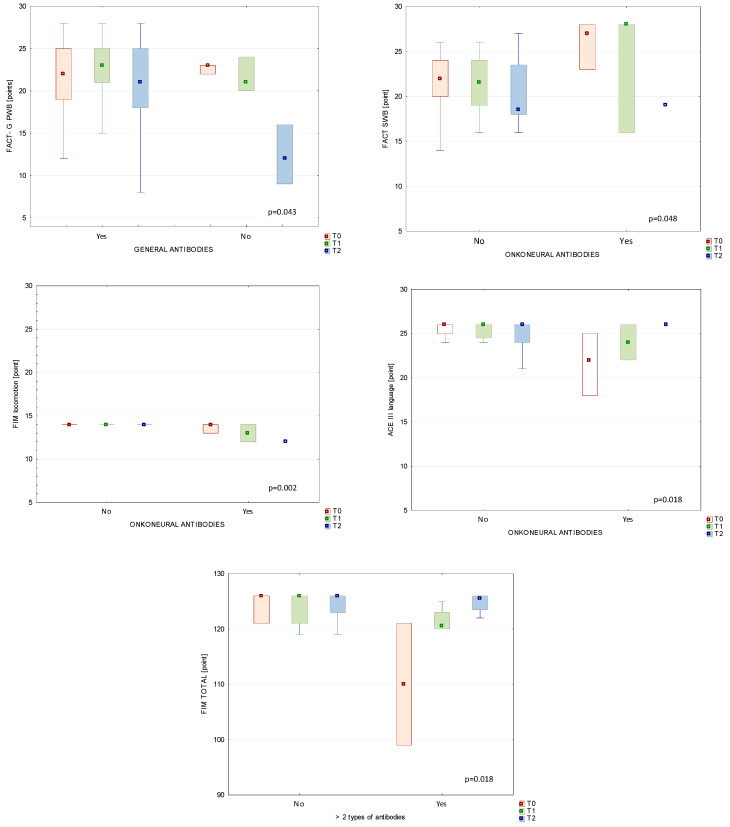
Functional subscores during study time between patients with autoantibodies and patients without humoral response.

**Table 1 diagnostics-14-00307-t001:** Characteristics of the study group.

Characteristic		Participants (*n* = 34), *n* (%) or Mean ± SD
Age	Years	48.80 ± 14.70
Years	Male	19 (55.9)
	Female	15 (44.1)
Education	Primary	0 (0.0)
	Vocational	8 (23.5)
	Secondary	17 (50.0)
	High	9 (26.5)
Type of tumour	Glioblastoma, grade 4	17 (50.0)
	Anaplastic astrocytoma	6 (17.6)
	Oligodendroglioma, grade 3	1 (2.9)
Total resection	Yes	18 (82.4)
	No	16 (17.6)
Tumour location, hemisphere	Right	20 (58.8)
	Left	14 (41.2)
Chemotherapy	Yes	29 (85.3)
	No	5 (14.7)
Steroids	Yes	34 (100.0)
	No	0 (0)

Abbreviations: SD, standard deviation; *n*, sample size.

**Table 2 diagnostics-14-00307-t002:** TJ proteins and S100β levels measured at baseline (T0), one day after RT (T1), and at three months post-RT (T2).

	T0	T1	T2	*p* Value
OCLN [pg/mL]	1.18	0.00	0.00	0.3454
median (IQR)	(0.0–5.96)	(0.00–7.57)	(0.00–3.94)	
CLN5 [pg/mL]	0.99	0.87	0.77	0.5857
median (IQR)	(0.37–2.48)	(0.24 –2.34)	(0.00–1.78)	
Zo-1 [RU/mL]	25.81	25.55	20.03	0.3944
median (IQR)	(12.51–50.74)	(13.47–50.83)	(11.99–48.14)	
S100β [pg/mL]	0.03	0.04	0.05	0.0531
median (IQR)	(0.01–0.63)	(0.01–0.2)	(0.02–0.14)	

Abbreviations: OCLN, occludin; CLN5, claudin 5; Zo-1, zonula occludens-1; S100β, S100 calcium-binding protein B; IQR, interquartile range.

**Table 3 diagnostics-14-00307-t003:** Functional status of study patients at baseline (T0), post-RT (T1), and at three months (T2).

Study Time Points	T0	T1	T2	*p* Value
	Median (IQR)			
Scale	FACT-G			
Total score	80.5 (46–107)	81 (43–108)	72 (43–103)	0.1482
PWB	23 (10–28)	24 (10–28)	21 (5–28)	0.1969
EWB	17 (4–24)	16 (7–24)	15 (6–24)	0.5982
SWB	24 (4–28)	24 (4–28)	23 (4–28)	0.0682
FWB	19 (10–28)	19 (6–28)	17 (6–27)	0.4429
Scale	ACE III			
Total score	88.4 (60–100)	88.82 (55–100)	87.7 (55–100)	0.6368
Attention	18 (8–18)	17.36 (13–18)	16.96 (11–18)	0.0043 *
Memory	21.8 (12–26)	21.5 (9–26)	21.3 (10–26)	0.5378
Language	24.9 (17–26)	24.7 (17–26)	24.9 (20–26)	0.3777
Fluency	10.2 (1–14)	10.18 (1–14)	10.4 (1–14)	0.1777
Visual–spatial	14.8 (8–17)	14.94 (6–17)	14.46 (8–16)	0.2709
Scale	FIM			
TOTAL score	124.8 (111–126)	124.1 (119–126)	118.3 (68–126)	0.0523
Self-care	41.7 (30–42)	41.9 (14–42)	39.5 (12–42)	0.0231 *
Mobility	21.1 (21–28)	21 (21–21)	19.5 (9–21)	0.3679
Locomotion	13.9 (12–14)	13.9 (12–14)	12.8 (6–14)	0.7165
Communication	13.9 (12–14)	13.7 (12–14)	12.7 (11–14)	0.0067 *
Sphincter Control	13.9 (12–14)	14.0 (14–14)	13.76 (10–14)	0.2231
Social Awareness	19.5 (14–21)	19.5 (15–21)	19.0 (10–21)	0.0498 *

Abbreviations: FACT-G, Functional Assessment of Cancer Therapy-General; PWB, Physical Well-Being (PWB); EWB, Emotional Well-Being; SWB, Social/Family Well-Being; FWB, Functional Well-Being; ACE III, Addenbrooke’s Cognitive Examination III; FIM, Functional Independence Measures; * *p* < 0.05.

## Data Availability

The datasets generated for this study are available on request to the corresponding author.

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
