# Peer review of "Neural and Onconeural Autoantibodies and Blood–Brain Barrier Disruption Markers in Patients Undergoing Radiotherapy for High-Grade Primary Brain Tumour"

_diagnostics, 2024, doi:10.3390/diagnostics14030307_

Round 1

Reviewer 1 Report

Comments and Suggestions for Authors

The proposed work is interesting and attempts to elucidate an interesting chapter within neurooncology. Two points caught my attention.

1-The study uses some serum biomarkers such as oclludin, claudin-5, Zo-1, S100 to asses the integrity of the BBB. I was not clear in the methodology why serum levels were assessed, especially the first 3, given that there are no robust studies.

2-What The authors talk about the relationship between the integrity of the BBB and these molecules, but these are histopathological studies, not serum studies. The best studies in this line are for the S100 beta biomarker.

 I compliment the authors for this hard and important work.

Comments on the Quality of English Language

No comments. It is for the MDPI.

Author Response

Dear Reviewer, 

Thank you very uch for your comments. We answer na your questions:

Q1:The study uses some serum biomarkers such as oclludin, claudin-5, Zo-1, S100 to asses the integrity of the BBB. I was not clear in the methodology why serum levels were assessed, especially the first 3, given that there are no robust studies.

Answ: The tight junction proteins are  important cell adhesion components that
stabilize endothelium cells lining and are responsible for maintaining the blood-brain barrier (BBB)integrity. For over 10 years, occludin, claudin-5 and ZO-1 are used as biomarkers of BBB damage in different pathologies i.e. during the early phases of stroke were they might be useful for predicting the risk of HT,in multiple sclerosis, brain tumors or fetal growth restriction syndrome. Such biomarkers could also facilitate the decision of whether to begin thrombolytic therapy in acute ischemic stroke patients, because the serum concentration reflects  BBB damage.

Q2: What The authors talk about the relationship between the integrity of the BBB and these molecules, but these are histopathological studies, not serum studies. The best studies in this line are for the S100 beta biomarker.

Answ: The studies on the clinical role of tight junction proteins (occludin, claudin-5 and ZO-1) are based not on histopathological analyses, but on serum levels.   In the "Discussion" section we have clarified that:

"Previous studies showed that not only at the tissue level, but also serum concentrations of TJs indicate BBB disruption and can be used as clinically useful biomarkers in a spectrum of central nervous system pathologies like ischemic stroke [45], fetal growth restriction syndrome [41,46,47] and multiple sclerosis [48]."

We have also additionally cited reports supporting such a statement.

Regards,

Authors

Reviewer 2 Report

Comments and Suggestions for Authors

So far, there is almost no effective therapy for brain tumor, even though it is better to have a radiotherapy, the negative impact functional outcomes some time may not be accepted due largely to the less selectivity to the normal tissue around or potential damage to the structure of the brain, especially the blood-brain-barrier. These authors evaluated the functional aspects and the markers related to BBB disruption through investigation of Neural and onconeural autoantibodies, and to verify the injury of blood-brain barrier in patients with primary brain tumor with radiotherapy. These clinical evidences definitely add additional sets of knowledge to the field to better understanding the value of radiotherapy. The aim, scientific question, the methods used, and conclusion are sound and clear. It should be acceptable after minor revision.

Minor concerns:

1. All figures should be modified for their clarity and they should be understandable easily without the description in the text, the marker for statistical significance should be included in the figures;

The units shown on veridical is too details/more than 10 and 5 is enough by increase the interval;

It would be OK to combine all 5 figure in one simply because the style of those figure is identical;

Figure legend should be inside of figure;

It would be great if all data points to be included with each bar figure alongside;

Sample size "n" should be described in the figure legend;

34 or 28 confused, please clarify;

2. The grammar, typo and consistency for the writing have to be addressed:

1) The possibility "P", capital or non capital;

2) "vs" should followed by ".';

3) Line 55: branches "o";

4) Line 60: indicates should be indicate corresponding to "levels";

5) Line 87:causes should be cause related to disruption and rise;

6) Line 131: There should be one space before "To";

7) Line 280: differed ..... from.......;

8) Line 381: bi-omarkers;

9) Reference: full page number should be given: #8, #45, #46;

10) Reference #37, #42, #48, the citation format is not correct;

Comments on the Quality of English Language

please see the comments

Author Response

Dear Reviewer,

Thank you very much for your comments.
We are very grateful for your comments and recommendations.

1. All figures should be modified for their clarity and they should be understandable easily without the description in the text, the marker for statistical significance should be included in the figures; The units shown on veridical is too details/more than 10 and 5 is enough by increase the interval; It would be OK to combine all 5 figure in one simply because the style of those figure is identical; Figure legend should be inside of figure; It would be great if all data points to be included with each bar figure alongside;

As requested by the review, we combined all 5 figure in one simply, unit scale were changed to 5th, P were added to charts. We hope it ok. Thank you.

2. The grammar, typo and consistency for the writing have to be addressed....

I will thank you for your suggestions, All recommended mistakes were changed.

Kind Regards,
Authors.